

# Numerical solutions of the flexure equation

David Hindle[1] and Olivier Besson[2]

[1]Georg-August-Universität Göttingen, Goldschmidtstr. 3, 37077 Göttingen, Germany
[2]Université de Neuchâtel, Rue Emile-Argand 11, CH-2000, Neuchâtel

**Correspondence:** David Hindle (dhindle@gwdg.de)

**Abstract.** The 4th order differential equation describing elastic flexure of the lithosphere is one of the cornerstones of geo-dynamics, key to understanding topography, gravity, glacial isostatic rebound, foreland basin evolution and a host of other phenomena. Despite being fully formulated in the 1940's, a number of significant issues concerning the basic equation have remained overlooked to this day. We first explain the different fundamental forms the equation can take and their difference in meaning and solution procedures. We then show how numerical solutions to flexure problems in general as they are currently formulated, are potentially unreliable in an unpredictable manner for cases where the coefficient of rigidity varies in space due to variations of the elastic thickness parameter. This is due to fundamental issues related to the numerical discretisation scheme employed. We demonstrate an alternative discretisation that is stable and accurate across the broadest conceivable range of conditions and variations of elastic thickness, and show how such a scheme can simulate conditions up to and including a completely broken lithosphere more usually modelled as an end loaded, single, continuous plate. Importantly, our scheme will allow breaks in plate interiors, allowing for instance, the creation of separate blocks of lithosphere which can also share the support of loads. The scheme we use has been known for many years, but remains rarely applied or discussed. We show that it is generally the most suitable finite difference discretisation of fourth order, elliptic equations of the kind describing many phenomena in elasticity, including the problem of bending of elastic beams. We compare the earlier discretisation scheme to the new one in 1 dimensional form, and also give the 2 dimensional discretisation based on the new scheme. We also describe a general issue concerning the numerical stability of any second order finite difference discretisation of a fourth order differential equation like that describing flexure where contrasting magnitudes of coefficients of different summed terms lead to round off problems which in turn destroy matrix positivity. We explain the use of 128 bit, floating point storage for variables to mitigate this issue.

## 1 Introduction

The elastic bending of the lithosphere under crustal loads is a fundamental part of modern geodynamics, describing a swathe of processes including glacial isostatic adjustment (Walcott, 1972), foreland basin formation in compression (Beaumont, 1981), and the flexural response of the lithosphere to extension (Egan, 1992). The mathematical theory was originally proposed in the pre-plate tectonic era by Gunn (1943a, b, 1947), who was interested in the wider question of compensation of loads by isostatic balance. Gunn wrote his series of papers at the culmination of a many decades long debate between geodesists and geologists concerning how loads on the crust and lithosphere were compensated by displacement of mantle material (cf. Barrell, 1914;



Gilbert, 1889; Putnam and Gilbert, 1895)). Geodesists had long favoured the idea that loads were all locally compensated (Airy or Pratt isostasy). Gunn was the first person to fully realise and formulate the necessary equations describing how a load can be compensated over a much greater distance than its own width due to the elastic strength of the lithosphere. Before him, it

was clear that several authors (Barrell, 1914; Gilbert, 1890) had had very similar insights but were never able to quantitatively demonstrate them (see Watts (2001) for a summary of the history of isostasy).

    Gunn's work transformed our understanding of lithospheric mechanics, establishing how loads on the crust were balanced by elastic bending of the lithosphere as well as suggesting how this compensation would affect measured gravity anomalies. Subsequently, in the early post-plate tectonic era, Walcott (1970a, c, b) published a series of manuscripts on the question of the

elastic thickness of the earth's lithosphere. Elastic thickness ($h$) is the key parameter in the flexural coefficient $D$ (the rigidity) which itself is related to the bending moment of the lithosphere (fig 1). A high value of $D$ means less bending of the lithosphere under loading. Following Walcott, estimating the value of elastic thickness $h$, especially for continental lithosphere became one of the most strongly debated topics in geodynamics (e.g., McKenzie and Fairhead, 1997; Watts, 2001; Audet and Mareschal, 2004; Kaban et al., 2018; Watts, 1992; Pérez-Gussinyé and Watts, 2005; Burov et al., 2006). At present, estimated values of $h$

for the continents range between 0 and 100 $km$.

    The fundamental equation describing the elastic bending of the lithosphere is our concern in this paper. Its original form presented to geologists by Gunn (1943b) remains unchanged. It describes the balance of forces (bending moment, vertical shearing forces, any added loads, and restoring forces from the buoyancy of mantle below) in an elastic plate resting on an (inviscid) fluid mantle, a so-called Winkler foundation (figure 1). Moment and shearing forces are converted to functions of the

deflection of the plate ($u(x)$ in this paper), resulting in a 4th order differential equation in $u$ given as

$$Du'''' + Pu'' - ku = q \qquad (1)$$

    where $u(x)$ is the vertical deflection of an originally horizontal surface of the plate due to loading, $u' = \dfrac{du}{dx}$ and so on for higher derivatives; $q(x)$ is a term describing the applied load; $P$ is a plate-wide stress; $k = (\rho_M - \rho_F) \cdot g$ is a constant allowing compensation of plate deflection by displacement of underlying "fluid" substrate (mantle) and $D = \dfrac{Eh^3}{12(1 - \nu^2)}$ is the flexural

rigidity in which $E$ is the elastic modulus, $h$ the elastic thickness of the lithosphere, $\nu$ poisson's ratio. For many solutions given in the literature $D$ is assumed constant, meaning $h$ does not vary along a plate's length. Should it be the case that $h(x)$ is variable however, then a modified general form of the equation above is

$$(Du'')'' + Pu'' - ku = q \qquad (2)$$

    Equations 1 and 2 can be reformulated in the following ways. We begin by rewriting explicitly to express things in terms of

forces and assuming $P = 0$.

$$Du'''' - (\rho_M - \rho_F)gu = \rho_L gq(x) \qquad (3)$$





We note that the term $(\rho_M - \rho_F)g$ corresponds to two forces: $\rho_M g$ is the restoring force due to displacement of mantle by deflection of the plate; $\rho_F g$ is a force exerted by material assumed to infill any surface deflections of the plate below an arbitrary reference level, but it is also clear that without modification, any solution of this problem equally assumes that infill

forces are *removed* wherever there is a positive deflection of the plate (figure 2). Such infill can range from nothing (empty basins) to water (oceanic cases for instance) to sedimentary product (foreland basins) or a mixture of any of the above in various combinations. We also note that the load term $\rho_L g q(x)$ defines a separate load of potentially different density to that assumed for the infill.

Two versions of the equation can be developed from here. The first one (figure 3a), which has rarely been explicitly discussed

or used in geodynamics, involves separating the infill load term from the mantle restoring force (see also appendix A).

$$Du'''' - \rho_M g u = \rho_L g q(x) - \rho_F g u \tag{4}$$

With the load term now entirely on the right hand side of the equation, we see it consists of two parts with the infill part, $\rho_F g u$ (right hand side), dependent on the deflection $u(x)$ for which we are solving. However, a fixed load $\rho_L g q(x)$, is also being applied which is itself an arbitrary function of $x$. The physical interpretation of this depends on how this fixed load term

is regarded. In most cases, it is probably assumed to be some kind of imposed crustal load such as a thrust sheet or ice sheet, or in oceanic cases, a seamount or volcanic island. Under these circumstances, it is equally clear that infill material cannot occupy space where the fixed load is applied, unless the special circumstance applies that the top of the fixed load at some point lies below the original reference level, in which case, a reduced accommodation space is available for infill material defined as the local sum $u(x) + q(x)$. In general, implementing a solution to this form of the problem requires numerical methods, since

arbitrary, piece-wise variations of load density may be required, and a solution will need to be iterative due to the dependence of the infill load term on $u(x)$. It is also important to state that the fixed load term $\rho_F g q(x)$ is actually an applied force scaled for a particular load thickness. Hence arbitrary "forces" can equally well be applied to the lithosphere, whatever their origin is assumed to be.

In general, it can be seen by examination that if $D = 0$ the solution will correspond to "Airy isostasy" (i.e. an "iceberg"

model) with

$$u(x) = \begin{cases} -\dfrac{\rho_L}{\rho_M} q(x) & q(x) \neq 0 \\ 0 & q(x) = 0 \end{cases} \tag{5}$$

An alternative development of the equation for solution is by division through by $(\rho_M - \rho_F)g$ giving

$$\frac{Du''''}{(\rho_M - \rho_F)g} - u = \frac{\rho_L}{(\rho_M - \rho_F)} q(x) \tag{6}$$





Here we see immediately that in contrast to equation 4, a direct, non-iterative solution is possible. This occurs because the physical meaning of the equation in this form is quite different. If we set $D = 0$ once more then

$$u(x) = \begin{cases} -\dfrac{\rho_L}{(\rho_M - \rho_F)} q(x) & q(x) \neq 0 \\ 0 & q(x) = 0 \end{cases} \tag{7}$$

Equation 7 also represents a case of Airy isostasy. However, in this form, $q(x)$ no longer represents a load thickness, but rather a load surface topography (figure3b), for which the appropriate flexural compensation function $u(x)$ is calculated. Hence, the difference $q(x) - u(x)$ gives the resulting finite load thickness (in figure 3b we explicitly show $Topo(x)$ as the "load" term and $q(x)$ as a term derived from $Topo(x) - u(x)$). As can be seen from equation 7 however, a problem with this formulation arises due to the different density terms employed, in particular the different values of $\rho_L$ and $\rho_F$. Only when $\rho_F = \rho_L$ is a condition of Airy isostatic balance of a load of density $\rho_L$ actually calculated, and hence the appropriate deflection $u(x)$ and resulting load thickness $q(x) - u(x)$. In analytical solutions of this equation, the only way to avoid this problem is by assuming $\rho_F = \rho_L$ everywhere. Otherwise, for piecewise variable density terms, a numerical solution is required.

In summary, the general equation of flexure of the lithosphere can be formulated in two different ways. In the first, a flexure dependent load term due to infill results, regardless of density variations, and requires an iterative solution due to the fact that regions with an imposed load cannot be simultaneously occupied by infill. Moreover, this form of the equation allows imposition of arbitrary forces to an elastic plate. A second and more common formulation of the problem describes the flexural subsidence required to support a particular surface topography. Analytical solutions to any flexure problem are generally unable to account for variable density of different load components, and to differentiate between fill of basins created by flexural subsidence and removal of fill in any positively deflected regions. Hence, we now consider some numerical solutions to flexure problems.

## 2 Numerical flexure solutions

For the most part, finite difference methods have been applied to solve the flexure equation numerically, both in the 1 dimensional beam type situation and for 2 dimensional, thin elastic sheets. For the simplest case of constant flexural rigidity, $D$, the left hand side of equation 3 for instance is discretised as (see figure 4)

$$\frac{D}{dx^4}(u_{i+2} - 4u_{i+1} + (6 - \frac{\rho_m g dx^4}{D})u_i - 4u_{i-1} + u_{i-2}) \tag{8}$$

Over the past 40 years, a number of numerical solutions to flexure problems were proposed. A lot of this effort was aimed at solving problems for the 2 dimensional extension of the flexure equation to an elastic sheet with variable elastic thickness and hence flexural rigidity (Van Wees and Cloetingh, 1994). Van Wees and Cloetingh (1994) had themselves corrected probably the earliest attempt at a numerical solution with variable flexural rigidity (Bodine et al., 1981). Relatively few publications have





dealt with the details of the numerical, 1 dimensional flexure equation discussed here, both for constant and variable elastic thickness cases. It should also be noted that since Van Wees and Cloetingh (1994) initial publication, most of the succeeding work involving numerical solutions of the flexure equation has been based on their numerical derivation (Stewart and Watts, 1997; Van der Meulen et al., 2000; Govers et al., 2009; Braun et al., 2013; Wickert, 2016).


An early paper using a numerical solution (Stewart and Watts, 1997) illustrates this form of the 1 dimensional numerical solution to the flexure equation with variable elastic thickness. Taking equation 2 for instance, the product rule of differentiation is applied prior to discretisation, hence

$$Du'''' + 2D'u''' + D''u'' - ku = q \qquad (9)$$

We note that this formulation of the problem, whilst mathematically correct, has a clear, unambiguous physical implication. Any function describing the variation of elastic thickness $h(x)$ (and hence rigidity, $D(x)$) as a function of position, must be continuous and at least twice differentiable. It will often be possible to get solutions to this problem with other functions of $D(x)$, but they are not consistent with the way the problem is posed in equation 9. The use of the product rule derivation of the problem extends to all of the aforementioned publications concerning the 2 dimensional sheet like problem as well (cf.

passing from equation 3 to equation 7 of Van Wees and Cloetingh, 1994). The resulting finite difference discretisation is given in appendix B

Perhaps surprisingly, there is an alternative and quite different method of discretising equation 2 available. This has been termed the "half station" method (Cyrus and Fulton, 1966, 1968), and avoids a product rule derivation prior to discretisation entirely. Instead, equation 2 is directly transformed into a finite difference approximation, replacing the derivatives both within

and outside the brackets with finite difference approximations to second derivatives (see appendix B). The main effect of this is that no third of fourth derivative terms in $u$ (and hence also first and second derivative terms in $D$) arise explicitly. Instead, the fourth order nature of the differential equation as well as gradients in $D$ are implicitly contained in the numerical scheme. Somewhat remarkably, this means that there is no restriction on the nature of the function $D(x)$. Any piecewise, abribtrary variation of $D(x)$ (and thus $h(x)$) is consistent with this discretised form of the equation. The half station method is

generalisable to 2 dimensions, meaning an alternative discretisation arises with equation 3 of Van Wees and Cloetingh (1994) as a starting point.

## 3  Comparison of the different numerical schemes

We now address the question of the different behaviours of the numerical schemes with respect to variations of $h(x)$. As we have already discussed, for the product rule version of the equation (so called "whole station" method (Cyrus and Fulton,

1968)) the formal derivation of the numerical scheme actually requires a continuous, at least twice differentiable function of $h(x)$. There are a number of interesting and illustrative cases of elastic thickness variation however where abrupt changes are required. We note that both whole and half station discretisations contain for any grid point or node $i$, terms involving $D_{i+1}, D_i,$



and $D_{i-1}$ (see figure 4 and appendix B) meaning that a discontinuity in $h(x)$ must be present across at least 3 grid nodes to take full effect. This was also commented upon by Wickert (2016) in the case of the whole station scheme, but without detailed

analysis of the results. A recognised difficulty with varying elastic thickness is how to find analytical solutions to which to compare numerical results. An indirect way of doing this however comes from Gunn (1943b), who showed analytically (cf. equations 13 and 16, Gunn, 1943b) that for the same, point load, the maximum deflection of a broken plate (i.e. one loaded at its end) is 4 times that of a continuous (infinite) plate equivalently loaded in its centre. Suppose we take a continuous plate and reduce the elastic thickness to zero over 3 nodes creating an elastic "break", and place a "point" load at a single node,

directly to the left or right of the elastic break. We note that the load in this case is not at a single point, but instead applied over a finite width equal to the grid spacing used in the numerical scheme, so direct comparison to an analytical solution is difficult. However, we would expect the plate, when loaded just next to the elastic break, to act like the end loaded or broken plate whereas the same plate loaded equally but without an elastic break should behave like the continuous one, so the relative maximum subsidence of the two numerical cases should change by a factor of 4. Results of the experiments are shown in

figure 5. The half station method gives exactly the result expected, showing that it corresponds to an end-loaded plate when elastic breaks or discontinuities are present within a larger plate. Due to the severe violation of the conditions of continuity and twice differentiability in the function of $h(x)$, the whole station method gives no meaningful result and is unable to simulate a "broken" plate.

Another interesting case to test the numerical schemes is that of what we can term an "isostatic raft". In this case, we simulate

an effectively infinitely stiff plate ($h \geq 500km$) with a central region, length $\sim 200km$, which is bordered at each end by an elastic break. The "raft" is loaded evenly, across its centreline by a rectangular shaped load of width $\sim 40km$. The relative dimensions are chosen purely to illustrate the point. If the plate segment were truly infinitely stiff, it would undergo no bending at all, and the load mass applied would be compensated by escape of an equal mass of mantle substrate. The half station method in this case produces an almost constant subsidence of the plate segment and causes a tiny amount of flexural bending. The

resulting mass difference between displaced mantle and load is $< 0.2\ \%$, showing the expected "raft-like" response. Again, for such a case, the whole station method produces a spurious result.

A geological application of the raft analogue arises when we consider tilted, crustal blocks formed in compression (McQueen and Beaumont, 1989). This refers to the concept of short segments of crust and lithosphere, bounded by basement transecting faults. McQueen and Beaumont initially created a simple force balance model where horizontal stress across a fault bounded

block generates a moment which turns and tilts the block against the resisting force of the mantle, and potentially augmented by the effects of erosion and sedimentation on the tilted block surface. This idea was principally used to explore the amount of compressive stress required to "break" the lithosphere in plate interiors, however, a corollary of it was to explain subsidence and basin formation as also due to the block tilting process. The model treats the block as completely rigid by default, and assumes the horizontal compressive force is responsible for the tilting, neglecting the effects of "self-loading" due to one block

overriding another and also flexural bending induced in the block.

Whilst our flexure model cannot be directly related to horizontal compressional stresses potentially involved in breaking the lithosphere, it is trivial to produce a succession of adjacent crustal segments by placing elastic breaks across a plate, creating





isolated segments of the desired dimensions. By loading each segment at or near its end and thus creating a turning force, and noting that any load can be treated as an arbitrary system of forces which arise for many different reasons, we produce a

similar result to that of McQueen and Beaumont but which also takes into account the flexural bending in the segments. Figure 7 shows two situations of identically loaded blocks with length 100 $km$ and 200 $km$. The longer blocks undergo substantially more bending, as a result of the mantle resisting force being spread over a greater length, and consequently holding the plate down more firmly, allowing it to bend elastically to a greater degree under loading. We note that a flexural model with a plate containing elastic breaks effectively parameterises lithospheric structure in terms of changes in elastic thickness. In the case

of the tilted block model, this parameterisation can be thought of as the net effect of bounding faults, with the applied load characterising any combination of the possible forces acting on a block (e.g. moment on the block due to horizontal stress, friction on the fault resisting tilting, self loading due to one block overriding another). In our illustrative models here, we have used 4 $km$ thick, 15 $km$ wide, distributed loads, corresponding to a net force of $1.6 \times 10^{12} N$. This produces relatively small uplifts at block corners, although clearly, the major component of uplift of block corners is likely to be due to steady

transport up basement faults bounding adjacent tilted blocks, due to shortening. The subsidence induced in basins by contast, can be more directly related to the response to loads on the crust. In the case of the Laramide orogeny for instance, maximum sedimentary thicknesses in the associated basins are $\sim$ 2 - 4 $km$ (Hagen et al., 1985) which is quite close to the $\sim 3 \ km$ subsidence under the load in our models.

All the preceding cases concern situations where the lithosphere is modelled as segmented or broken. In many cases however,

we consider elastic thickness to vary more steadily (e.g., Van der Meulen et al., 2000; Stewart and Watts, 1997). In such cases, the most straightforward spatial variation of $h(x)$ is described by a piecewise, linear function, interpolated between a few points of fixed value. We find that in cases where the gradient of the imposed linear change is not too sharp, the whole station approximation can return reasonable results, including in cases where $h \to 0$. Below a certain threshold however, the error (which we take as the difference to the half station solution), quickly reaches $\geq 2\%$ which for studies fitting flexural curves to

gravity signals for instance (e.g., Stewart and Watts, 1997), will be critical. Figure 8 shows two cases, with differing gradients in $h(x)$. While the gentle gradient yields a difference of 6 $m$ maximum subsidence ($\sim 0.5\%$) the sharper gradient reaches 340 $m$ ($\sim 30\%$). As the gradient in $h(x)$ increases still further, the whole station scheme will ultimately reach a point where it returns no solution at all whilst the half station method is stable for any combination of loads and variations of $h(x)$.

## 4 Other numerical issues

A more general mathematical issue concerns the positivity of any numerical solution to a fourth order differential equation of this kind. As can be verified (see appendix B), all discretisations of the flexure equation using second order finite difference approximations will yield an identical set of linear equations when the value of $h$ is a constant. The discretised form then becomes that shown in equation 8. It can also be seen that for constant values of $D$ at least, for each line, the sum across the columns ($\sum_j a_{i,j}$) is $\rho_M g$, the mantle restoring force, since all other terms involving $D$ sum to zero. As a result, the residual

term due to the mantle restoring force is, according to the maximum principle (Axelsson, 1994), necessary for maintaining the





positivity of the system of equations represented by equation 8. Hence, the composition of the main diagonal, which consists itself of a sum of two terms, $\frac{6D}{dx^4}$ and $\rho_M g$ becomes of critical importance. This is due to the issue of round off, where the capacity of 64 bit representations of numbers to sum terms with large contrasts in magnitude leads to the smaller term being partly or entirely lost as the maximum number of significant figures available in arithmetic operations (approximately 15) is

exceeded. In the large term, $D$ will vary as a function of $h^3$ where elastic thickness $h$ may reach values of $100 \ km$ on Earth (Kaban et al., 2018), and possibly even $300 \ km$ on Mars (Thor, 2016). Grid spacing $dx$ requires values of $\sim 100 \ m$ or less to ensure convergence, and also to allow resonable resolution in the representation of loads. It should be noted that currently, the highest resolution, public and globally available topographic databses, SRTM and ASTER GDEM, are both on 1 arc second ($\sim 30m$) grids whilst TANDEM-X data, accessible relatively freely to scientists is on an 0.4 arcsec ($\sim 10 - 12m$) grid. Hence,

$\frac{6D}{dx^4}$ will quite conceivably be of magnitude $10^{19}$ or more in "real" problems in geodynamics. The small term $\rho_M g$ will always be of order $10^4$. Under such conditions, round off will lead to an "effectively singular" matrix and the numerical problem will fail. One work around is relatively easily available. The use of quadruple precision (128 bit) representation of floating point numbers allows $> 30$ significant figures to be taken account of in arithmetic operations. Although this costs additional memory and some speed, it assures that any conceivable problem of flexure with "real world" dimensions and parameters will

be correctly dealt with by the numerical algorithm.

Besides the general issue described above, the additional term for the plate wide stress $P$ (see equation 2 and appendix B) also has the potential to cause problems for the numerical solution. In particular, $P$, which has always been assumed constant throughout a plate, will interact with regions of variable elastic thickness in a potentially problematic way. For cases with elastic breaks, where $D \to 0$ for instance, a constant value of $P$ will often lead to failure of the numerical solution. In such

cases, it is probably reasonable to set $P \to 0$ on the three nodes of the discontinuity, thereby treating these as if they were an infinitely thin fault. Because the nodes either side of the break have the normal value of $P$ applied, the continuity of $P$ is to some degree respected.

## 5   Implications

The half station method of discretisation we have presented here is clearly able to deal with a complete set of possible variations

of elastic thickness in the flexure equation. The whole station method (applying the product rule first) by contrast, is unable to be relied upon to do so. Perhaps the most worrying aspect of the whole station method is that in some circumstances, it will give results that appear plausible, but are in fact in error by significant amounts (fig 8b), and actually represent solutions among the set of transitional variations of $h$ just before the method fails completely. The principal reason for the behaviour of the whole station discretisation is the fact that as posed, a condition of the equation is a twice differentiable, continuous function

of $D$. Were we to apply such functions, the whole station discretisation would perform safely. However, it is also known that the half station method works with smoothly varying functions of $D$, and *in general*, the errors associated with the half station method are always smaller than those of the whole station (Cyrus and Fulton, 1968). It is also unjustifiable to impose any





such constraint on the nature of variations of elastic thickness of the lithosphere. Consequently, it appears clear that the finite difference discretisation of the flexure equation should be carried out using the half station method.

The wider application of the half station method to many other differential equations with variable coefficients was originally noted by Cyrus and Fulton (1966, 1968). The specific application of it to the 1 dimensional wave equation and the general rarity of its use has also been discussed by Langtangen, (p. 44). It seems clear that this form of finite difference discretisation which allows arbitrary, piecewise variations of coefficients is a potentially significant and generally overlooked method for the wider spectrum of the physical sciences.

The use of numerical solutions for the flexure equation covers many aspects of geodynamics. On the one hand, the determination of the elastic thickness of the lithopshere can be done using a forward modelling approach with flexure models used to match gravity data, and topography (e.g., Walcott, 1970a; Karner and Watts, 1983; Watts, 1992; Stewart and Watts, 1997). In such cases it may well be necessary to look for solutions incorporating variable elastic thickness, especially around mountain fronts in foreland basins. Flexure models may also be used to study the dynamics of past flexural events (Burkhard and
Sommaruga, 1998; DeCelles and Giles, 1996; Horton and DeCelles, 1997; Beaumont, 1981; Hagen et al., 1985; Hindle and Kley, 2020) where they are often used to model subsidence patterns and explain basin formation. In many of these cases too, variable elastic thickness is likely to need taking account of. Increasingly, topics relating to global sea-level rise and the melting of the polar ice caps will demand high resolution models of flexural responses which may require taking account of changes in elastic thckness of the lithosphere. More generally, the issue of elastic breaks within continental lithosphere has yet to be
substantially explored, and could have signifcant consequences for topics such as intraplate seismicity and seismic hazard. In short, it seems very important to make such numerical approximations in as accurate a way as possible. Current flexure models (Wickert, 2016) are based on the whole station (product rule) derivation of the numerical scheme, and should be revised.

## 6   Conclusions

Despite a long history of use in the literature, and an apparent sense of being work completed, in fact a host of problems arising
from a simple numerical analysis of the discretised flexure equation have remained untouched. When we examine these, we find there are significant issues with the method of discretisation used. It is not advisable under any circumstances to use a product rule derivation of an equation of this type when the coefficient varies as a function of coordinate. Realistic models of natural variations in elastic thickness (and many other coefficients in many other equations arising in natural sciences in general), will require sharp changes in those coefficients to be taken account of. A product rule scheme cannot do this successfully, especially
for fourth order differential equations.

    Fundamental problems relating to the nature of the system of linear equations arising from discretisations of the flexure equation have also gone unnoticed so far. For small grid spacings, something which will inevitably become increasingly common as computer power increases, the numerical solution will rapidly become unstable and fail, unless a 128 bit floating point representation is employed. If this is used, the problem will probably be avoided at grid spacings $\geq 1m$, but below this
threshold, instability could once more arise quite easily. Although we have presented only 1 dimensional problems in this





paper, it is nevertheless clear that everything shown here extends to 2 dimensional, thin elastic plate formulations as well. To this end, we give the 2 dimensional half station formulation and discretisation of the problem (appendix B). We will discuss 2 dimensional solutions using this scheme in forthcoming papers.

*Code availability.* The codes used in the preparation of this paper are available from the github repository, Hindle (2021)

**Appendix A: General aspects of the equation when solved numerically**

We begin with a general formulation of the flexure problem with variable coefficient and specified load (not topography) as follows. This form of the problem requires an iterative solution. We give the form of the equation for the case where the gravitational constant is negative, i.e. $g = -9.81$, meaning load thickness $q(x) > 0$ acts as a downwards force on the lithosphere.

$$(D(x)u'')'' + Pu'' - \rho_M gu = \rho_L gq(x) - \rho_F gu \tag{A1}$$

$u$ is the deflection of the plate at position $x$, along its length. $D(x)$ (the flexural rigidity) varies in space and is explicitly written as a function of $x$. The value of $D(x)$ is given by $Eh(x)^3/(12 \cdot (1 - \nu^2))$, where $E$ is the elastic modulus of the lithosphere, $h(x)$ is the effective elastic thickness of the lithosphere and is the parameter in $D(x)$ that varies in space, and $\nu$ is Poisson's ratio. $P$ is a constant representing a plate wide, horizontal stress. The term $\rho_M \cdot g$ (left hand side) represents a
restoring force due to displaced mantle. On the right hand side of the equation, $\rho_L gq(x)$ is the imposed load term which is chosen arbitrarily and has a density $\rho_L$ that can be set for whatever load is being modelled. However, this load term can also be thought of as representing any type of force loading the plate (for instance forces across a fault resolved in the vertical direction or torques from horizontal loads on rigid blocks). $\rho_F gu$ is the load force due to "infill" of basins. However, due to its dependence on $u$, the term acts as a load when $u$ is negative and a positive force (pushing or pulling the plate upwards) when
$u$ is positive. This upward pull can be thought of as a force due to erosional removal of material and by default, the amount of erosion is equal to the value of $u$, as if the uplifted segment of plate were eroded to zero metres above reference level (fig 2, main text). We may wish to make the density of eroded material different to that of infill, for instance a "crustal" density $\rho_C$. In this case, the full equation is dependent on the sign of $u(x)$ and can be written as

$$(D(x)u'')'' + Pu'' - \rho_M gu = \begin{cases} \rho_L gq(x) - \rho_F gu & u(x) < 0 \\ \rho_L gq(x) - \rho_C gu & u(x) \geq 0 \end{cases} \tag{A2}$$

Equally, basin fill is assumed to fill basins completely to the same reference level. We note that different values for erosion and fill levels (even spatially variable and piecewise) can be implemented relatively easily with a numerical code. For analytical solutions to the problem, it is implicit that there is infill and erosion and that density of all materials is the same. We also state





again that the first aim of the iterative scheme is to separate regions filled with fixed load, where there is subsidence given by $u(x)$ but clearly, no accommodation space exist for infill, from the basins created outside the regions occupied by the load.

With a numerical method, it is relatively easy to assure this is the case.

## Appendix B: Discretisation schemes

### B1  Half station discretisation

We apply second order finite difference operators simultaneously for both second derivatives, inside and outside the brackets in equation A1, something referred to as the half station method (Cyrus and Fulton, 1968).

Hence, if

$$f'' \approx \delta^2 f = (f_{i+1} - 2f_i + f_{i-1})/dx^2 \tag{B1}$$

where $dx$ is the grid spacing, and $i$ is the node number, then

$$(Du'')'' \approx \delta^2(D\delta^2 u) \tag{B2}$$

We discretise the whole term in brackets first, on a grid $i = 1, ..., N$

$$\delta^2(D\delta^2 u) = ((D\delta^2 u)_{i+1} - 2(D\delta^2 u)_i + (D\delta^2 u)_{i-1})/dx^2 \tag{B3}$$

then, substituting the terms in brackets and advancing the indices

$$\delta^2(D\delta^2 u) = ((D_{i+1}(u_{i+2} - 2u_{i+1} + u_i)) - 2(D_i(u_{i+1} - 2u_i + u_{i-1})) + (D_{i-1}(u_i - 2u_{i-1} + u_{i-2}))/dx^4 \tag{B4}$$

collecting terms, we obtain

$$\delta^2(D\delta^2 u) = (D_{i+1}u_{i+2} - 2(D_{i+1} + D_i)u_{i+1} + (D_{i+1} + 4D_i + D_{i-1})u_i - 2(D_{i-1} + D_i)u_{i-1} + D_{i-1}u_{i-2})/dx^4 \tag{B5}$$





discretising the remaining parts of the equation then gives

$$\frac{D_{i+1}}{dx^4}u_{i+2}$$

$$+\left(-2\frac{(D_{i+1}+D_i)}{dx^4}+\frac{P_i}{dx^2}\right)u_{i+1}$$

$$+\left(\frac{(D_{i+1}+4D_i+D_{i-1})}{dx^4}-2\frac{P_i}{dx^2}-\rho_M\cdot g\right)u_i \tag{B6}$$

$$+\left(-2\frac{(D_{i-1}+D_i)}{dx^4}+\frac{P_i}{dx^2}\right)u_{i-1}$$

$$\frac{D_{i-1}}{dx^4}u_{i-2}$$

$$=(q_o)_i+q(u_i)$$

where the two load terms, $(q_o)_i$ and $q(u_i)$ represent the static, fixed load and the iteratively calculated infill load respectively. If we gather all coefficients into a matrix $\mathbf{A}$ and form a matrix equation, the resulting system is of the form

$$\mathbf{A}u=q(u) \tag{B7}$$

a non-linear series of equations in $u$. We reformulate this as a recursive matrix fixed point problem, which we solve using a pentadiagonal matrix algorithm (Sebben and Baliga, 1995)

   A similar procedure is used to discretise the specified topography formulation (equation 3)

**B2   Whole station discretisation**

The whole station discretisaton begins from the result of applying the product rule to B1 giving us

$Du''''+2D'u'''+D''u''+Pu''-\rho_Mgu=\rho_Lgq(x)-\rho_Fgu$         (B8)

   Discretisation involves applying second order finite difference schemes directly to all derivatives.





Hence

$$Du'''' \approx \frac{D_i}{dx^4}(u_{i+2} - 4u_{i+1} + 6u_i - 4u_{i-1} + u_{i-2})$$

$$2D'u''' \approx \frac{(D_{i+1} - D_{i-1})}{2 \cdot dx^4}(u_{i+2} - 2u_{i+1} + 2u_{i-1} - u_{i-2})$$  (B9)

$$D''u'' \approx \frac{(D_{i+1} - 2D_i + D_{i-1})}{dx^4}(u_{i+1} - 2u_i + u_{i-1})$$

Adding the remaining terms from equation B8 and writing explicitly to show the relationship to B9, we have

$$\left(\frac{D_i}{dx^4} + \frac{(D_{i+1} - D_{i-1})}{2 \cdot dx^4}\right) u_{i+2}$$

$$+ \left(-4\frac{D_i}{dx^4} - \frac{(D_{i+1} - D_{i-1})}{dx^4} + \frac{(D_{i+1} - 2D_i + D_{i-1})}{dx^4} + \frac{P_i}{dx^2}\right) u_{i+1}$$

$$+ \left(6\frac{D_i}{dx^4} - 2\frac{(D_{i+1} - 2D_i + D_{i-1})}{dx^4} - 2\frac{P_i}{dx^2} - \rho_M \cdot g\right) u_i$$  (B10)


$$+ \left(-4\frac{D_i}{dx^4} + \frac{(D_{i+1} - D_{i-1})}{dx^4} + \frac{(D_{i+1} - 2D_i + D_{i-1})}{dx^4} + \frac{P_i}{dx^2}\right) u_{i-1}$$

$$+ \left(\frac{D_i}{dx^4} - \frac{(D_{i+1} - D_{i-1})}{2 \cdot dx^4}\right) u_{i-2}$$

$$= (q_o)_i + q(u_i)$$

The same procedure as for the half station discretisation is employed to solve these equations. The iteration could be made more efficient.

## B3 Half station, 2 dimensional discretisation

As has been discussed, all 2 dimensional, thin elastic sheet type solutions used up to the present day have been based on the
same product rule derivation of the numerical scheme (whole station). It is equally possible to apply a half station derivation however. We start from equation 3 of Van Wees and Cloetingh (1994) and assuming $\nu$ constant, we can write

$$(Du_{xx})_{xx} + (Du_{yy})_{yy} + \nu((Du_{xx})_{yy} + (Du_{yy})_{xx}) + 2(1-\nu)(Du_{xy})_{xy} + P\Delta u - \rho_M gu = \rho_L gq(x) - \rho_F gu$$  (B11)





where $u_{xx} = \dfrac{\partial^2 u(x,y)}{\partial x^2}$ and so on. Defining partial finite difference operators on a 2 dimensional grid, $i,j$

$$\frac{\partial f}{\partial x} \approx \frac{1}{dx}(f_{i+1/2} - f_{i-1/2})$$

$$\frac{\partial f}{\partial y} \approx \frac{1}{dy}(f_{j+1/2} - f_{j-1/2})$$

(B12)

and assuming $\lambda := dx = dy$, equation B11 can be discretised term by term, in a similar way to the half station method applied to the 1 dimensional form. We have used Maxima to derive the solution.

$$
\begin{aligned}
&u_{i+2,j}\, 2D_{i+1,j} + \\
&u_{i+1,j+1}\, [-(\nu-1)D_{i+1,j+1} + (\nu+1)D_{i+1,j} + (\nu+1)D_{i,j+1} - (\nu-1)D_{i,j}] + \\
&u_{i+1,j}\, [(\nu-1)D_{i+1,j+1} - 2(\nu+3)D_{i+1,j} + (\nu-1)D_{i+1,j-1} + (\nu-1)D_{i,j+1} - 2(\nu+3)D_{i,j} + (\nu-1)D_{i,j-1} + 2\lambda^2 P_{i,j}] + \\
&u_{i,j+1}\, [(\nu-1)D_{i+1,j+1} + (\nu-1)D_{i+1,j} - 2(\nu+3)D_{i,j+1} - 2(\nu+3)D_{i,j} + (\nu-1)D_{i-1,j+1} + (\nu-1)D_{i-1,j} + 2\lambda^2 P_{i,j}] + \\
&u_{i,j}\, [-(\nu-1)D_{i+1,j+1} - 2(\nu-2)D_{i+1,j} - (\nu-1)D_{i+1,j-1} - 2(\nu-2)D_{i,j+1} + 4(3\nu+5)D_{i,j} - 2(\nu-2)D_{i,j-1} \\
&\qquad\quad -(\nu-1)D_{i-1,j+1} - 2(\nu-2)D_{i-1,j} - (\nu-1)D_{i-1,j-1} - 8\lambda^2 P_{i,j} - 2\lambda^4 \rho_M g] + \\
&u_{i+1,j-1}\, [(\nu+1)D_{i+1,j} - (\nu-1)D_{i+1,j-1} - (\nu-1)D_{i,j} + (\nu+1)D_{i,j-1}] + \\
&u_{i,j-1}\, [(\nu-1)D_{i+1,j} + (\nu-1)D_{i+1,j-1} - 2(\nu+3)D_{i,j} - 2(\nu+3)D_{i,j-1} + (\nu-1)D_{i-1,j} + (\nu-1)D_{i-1,j-1} + 2\lambda^2 P_{i,j}] + \\
&u_{i,j+2}\, 2D_{i,j+1} + \\
&u_{i-1,j+1}\, [(\nu+1)D_{i,j+1} - (\nu-1)D_{i,j} - (\nu-1)D_{i-1,j+1} + (\nu+1)D_{i-1,j}] + \\
&u_{i-1,j}\, [(\nu-1)D_{i,j+1} - 2(\nu+3)D_{i,j} + (\nu-1)D_{i,j-1} + (\nu-1)D_{i-1,j+1} - 2(\nu+3)D_{i-1,j} + (\nu-1)D_{i-1,j-1} + 2\lambda^2 P_{i,j}] + \\
&u_{i-1,j-1}\, [-(\nu-1)D_{i,j} + (\nu+1)D_{i,j-1} + (\nu+1)D_{i-1,j} - (\nu-1)D_{i-1,j-1}] + \\
&u_{i,j-2}\, 2D_{i,j-1} + \\
&u_{i-2,j}\, 2D_{i-1,j} \\
&= 2\lambda^4 [(q_o)_{i,j} + q(u_{i,j})]
\end{aligned}
$$

(B13)

*Author contributions.* David Hindle had the idea to use the half station method of discretisation (without at the time realising he was doing so and by complete accident). Olivier Besson pointed out the implications of the discretisation, discovered the earlier work on it, and also
explained the reasons behind the instability of the numerical method when round off becomes an issue. Olivier Besson modified the code to work with 128 bit, quadruple precision, floating point representation. Olivier Besson developed a number of benchmark codes to check results against original papers. Olivier Besson wrote the maxima script to get the correct half station derivation of the 2 dimensional problem. The main codes used in the paper were written by David Hindle. The main text, figures and experiments were all prepared by David Hindle.



*Competing interests.* There are no competing interests in this research.

*Acknowledgements.* David Hindle thanks Hans Petter Langtangen posthumously for his enormous contributions to the process of learning and understanding mathematics for natural scientists. David Hindle only discovered Hans Petter's recognition of the half station method amongst the many, rich web archives of course material Hans Petter left to us all, long after Hans Petter Langtangen's sad and early death. This is a testament to the enduring influence of an evidently much loved, talented and inspiring individual, and his wonderful scientific career which was cut far too short.





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





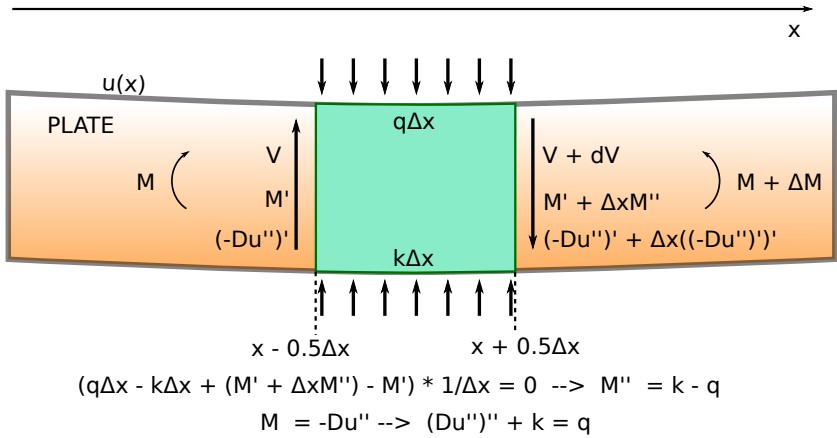

**Figure 1.** The force balance across a segment of a plate, $\Delta x$ and the derivation of the fourth order differential equation describing elastic flexure. The segment is supported from below by mantle restoring forces ($k\Delta x$) and loaded from above by a distributed load $q\Delta x$. Shear stress $V$ is equal to the first derivative of the Moment $M'$, which in turn is equal to the coefficient $D$ multiplied by the second derivative of deflection of the plate $u''$. Over the small distance, between $x - 0.5\Delta x$ and $x + 0.5\Delta x$, the flexural bending equation given above arises.


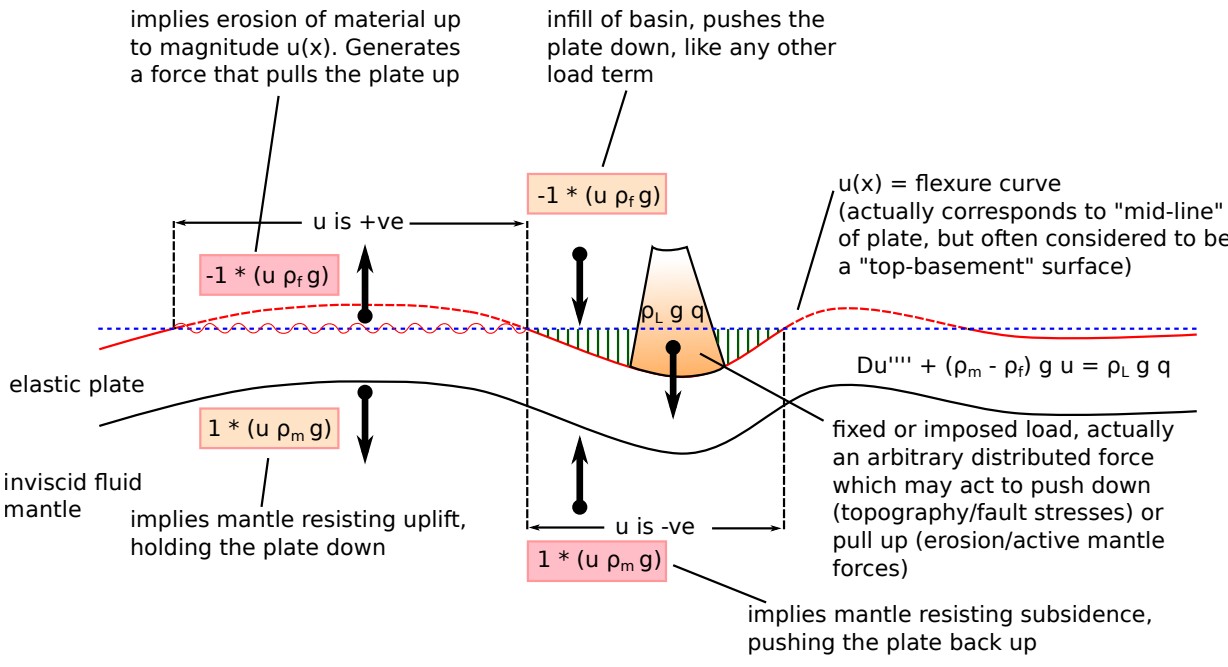

**Figure 2.** Flexure equation and its physical significance. "Restoring" forces (constant multiplied by the plate vertical deflection, $u(x)$) have differing effects according to whether $u(x)$ is positive when there is, by default, erosion of the plate to level zero, or when negative, there may be infill of basins created by flexural subsidence. Mantle forces are always present, and damp subsidence due to surface loading, but equally damp uplift when flexure bends things above zero reference level.





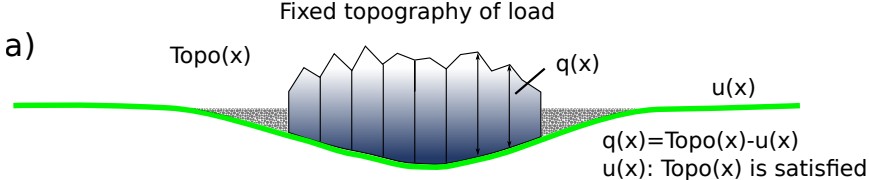

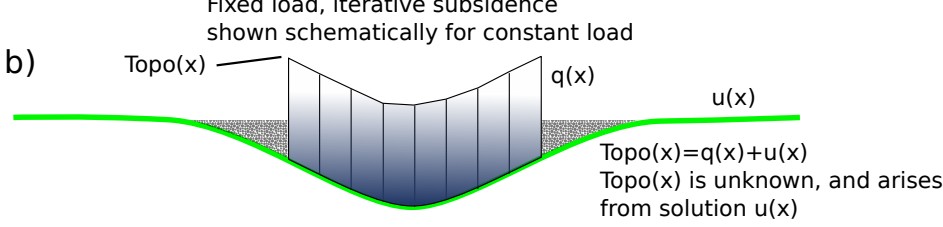

**Figure 3.** The two types of flexure model derived from equation 2. a) the "fixed topography" situation where subsidence matches the prescribed topographic profile (associated with load density). Load thickness is then equal to topography $Topo(x)$ minus subsidence $u(x)$. b) the fixed load case, equivalent to arbitrary forces. Any "load" may be applied generating subsidence. To calculate infill of basins generated, should there be any, requires an iterative procedure since the amount of accommodation space must be first calculated explicitly and subsequently filled, hence creating more accommodation space.





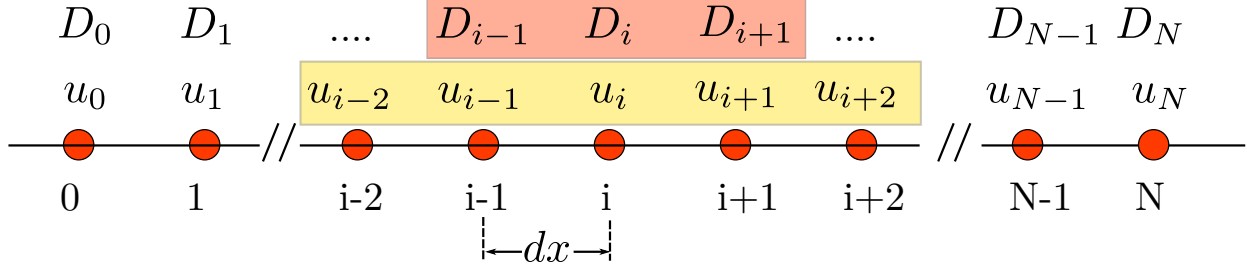

**Figure 4.** Grid and finite difference stencil for solution of flexure equation with variable elastic thickness, showing the grid elements involved in the discretisation of the problem pertaining to the value of $u_i$ at grid node $i$. Variation in $D$ (due to variation in $h$) at any node $i$ is achieved across grid nodes $i-1, i, i+1$. Hence, for abrupt changes in $D$ and $h$, the value of $h$ must be adjusted for at least 3 adjacent nodes in order to take full effect. The discretisation of the derivative of $u$ requires the five nodes $i-2, ..., i+2$. Grid spacing $dx$ gives the problem a physical dimension.



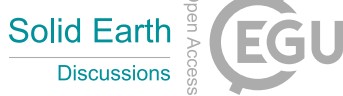

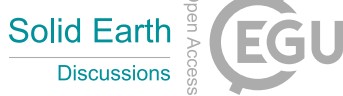

**Figure 5.** Comparison of broken and unbroken plate loading simulated with "line load" (for parameters used, see Table A1) a) "broken" plate contains 3 node break of $h = 0.01m$ b) continuous plate, same load. Maximum subsidence in ratio $\sim 4{:}1$ with small discrepancy due to the fact that the load is not a true "line" load but rather, has a finite width equal to the grid spacing. This corresponds exactly to the analytical results of Gunn (1943b). Figure prepared using GMT v6.0.0 (Wessel et al., 2019).





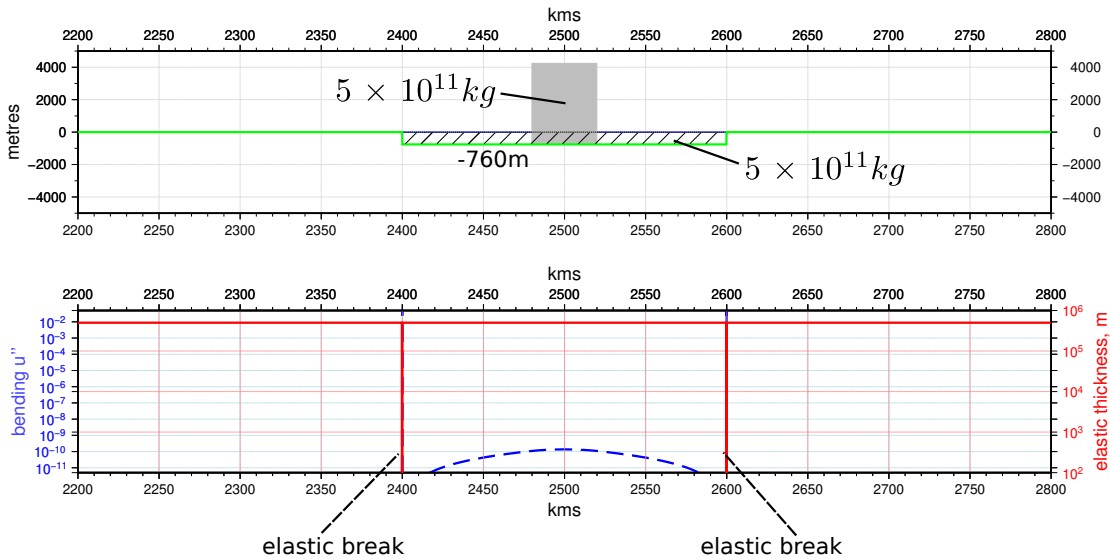

**Figure 6.** Isostatic "raft" model. With an effectively infinite thickness plate and a symmetrically loaded "raft" which is detached at both ends by elastic breaks ($h = 0.01m$ for 3 nodes) the mass of mantle displaced ($760\,m$ thick layer) is almost exactly equal to the mass of the applied load, demonstrating isostatic balance without flexure (for parameters used, see Table A1). Figure prepared using GMT v6.0.0 (Wessel et al., 2019).



**Figure 7.** Tilted block model directly equivalent to force balance model of McQueen and Beaumont but also incorporating flexure. Calculations assume basins filled with infill material (red shading) and plate surface is eroded to zero topography, removing material (blue shading). For parameters used, see Table A1. a) 100 $km$ block length. Blocks are visually close to rigid and tilted b) 200$km$ blocks with identical loading which undergo substantial bending ($\sim 5$ times that of the 100 $km$ block). Elastic bending increases as the plate is more strongly held down by the greater length over which mantle resistance forces can act. It is important to note however, that the plate segments have unconstrained boundaries and are held in place only by their interaction with the mantle. Figure prepared using GMT v6.0.0 (Wessel et al., 2019).





**Figure 8.** Piecewise linear variation of $h(x)$, (for parameters used, see Table A1) a) relatively gentle gradient, where whole station and half station schemes are in good agreement. b) sharper gradient where there is a substantial difference between whole station and half station. Figure prepared using GMT v6.0.0 (Wessel et al., 2019).



**Table A1.** Parameters used for figs 5-8

| Parameter | Fig 5 | Fig 6 | Fig 7 | Fig8 |
|---|---|---|---|---|
| Background elastic thickness $km$ | 20 | 500 | 20 | 30 |
| Weak zone minimum elastic thickness $km$ | 0.001 (5a) | 0.01 | 0.01 | 15 |
| Poisson's ratio $\nu$ | 0.25 | " | " | " |
| Plate-wide stress $Nm^{-1}$ | 0 | " | " | " |
| Static load density $kgm^{-3}$ | 2700 | 2500 | 2700 | 2700 |
| Infill load density | 0 | 0 | 2300 | 2300 |
| Crustal density | 0 | 0 | 2700 | 2700 |
| Mantle density | 3300 | " | " | " |
| Grid spacing $m$ | 100 | " | " | " |
| Nodes | 50001 | " | " | " |