# Peer review of "A corrected finite difference scheme for the flexure equation with abrupt changes in coefficient"

_Solid Earth, 2021_

## Author Response (AR1)

Please note, we have already made point-by-point responses to the reviewer's comments. We nevertheless highlight here all changes made and documented by the diff.tex and diff.pdf files created using latexdiff on the original and revised manuscripts. Line nos. refer to the file diff.pdf

1)  We have changed the title to reflect the content and objectives of the article much more accurately, as suggested by reviewer 2.

2) line 25: We examined the manuscript of Smoluchowski (1909) and noted that it was concerned uniquely with buckling due to horizontal force. The more widely accepted and used elastic theory, which is what this paper is concerned with, is the compensation of vertically acting crustal loads by the elastic lithosphere. The origins of this theory are traced back to Bernouilli and Euler in the 1700's although it is very difficult to cite a specific work in this context. Hence, we mention their contributions to the theory as being its origin, but it remains the case that Gunn was the first person to formulate a "Winkler" foundation model using a hydrostatic compensation term, and dealing with vertical loads.

3) line 42: We have added a sentence clarifying why there is so much variation in the determined values of effective elastic thickness.

4) line 53: we have clarified the term P (horizonal compressive force) in the equation.

5) line 60 onwards – the units issue is related to body forces – the volume integral of the density or density difference of a body/bodies multiplied by the gravitiational acceleration – the units are correct since it is a unit volume integral (m*m*m = m3) over unit vertical columns, divided by density (m-3) – yielding ultimately Newtons for units. Hence, we have changed the text to state "body force" and also explicitly stated "per unit length/width with units of Newtons).

6) line 59-60: we have changed the text to clarify "things".

7) line 83: we have explained why there is no infill in a case of perfect Airy isostacy (D=0)

8) line 142: we clarify the proof of convergence given for the half and whole station methods in general in Cyrus and Fulton. We also point out that this is itself a limited case of the Lax-Milgram theorem, the underpinning theorem of all grid based, discrete numerical approximation methods for solving pde's. We reference the newly added appendix C which contains various proofs of the convergence problems of the whole station method in particular for certain cases of abrupt changes in coefficient (this proof is more widely applicable in truth). We note here only that the reviewers' assertion (not a conjecture, and with no proof offered) is actually contradicted in certain cases of the proofs given in appendix C. There is no way to get a "smooth" equivalent to an abrupt variation of coefficient, that is compatible with a whole station approximation. Even if there were, it would be limited by the rounding error/numerical precision problem we have demonstrated in the paper.

9) Line 360 onwards: Appendix C and additional figure – we give proofs of convergence for the 4$^{th}$ order de by application of Lax-Milgram, including the limitation to the domain of a Sobolev space (this goes further than Cyrus and Fulton's papers from the 1960's). We show the effect of regularity of D on the whole station method.  We give a comparison of a 2$^{nd}$ order, pde on a 2d domain solved using a finite element method, which is roughly equivalent to the 4$^{th}$ order de used for the flexure equation in this paper. The difference for a case with abrupt variation in coefficient is small. This will always be the case within the limits (small strain approximation) of both equations.

---

## Author Response (AR2)

All changes requested by reviewer 2 were made, except that we could not find a double mention of Cyrus and Fulton (line 381). This may have been a LaTeX error which self-corrected.

---

## Author Response (AR3)

All changes requested by reviewer 2 were made, except that we could not find a double mention of Cyrus and Fulton (line 381). This may have been a LaTeX error which self-corrected.